

# Phospho-islands and the evolution of phosphorylated amino acids in mammals

Mikhail Moldovan[1] and Mikhail S. Gelfand[1,2]

[1] Skolkovo Institute of Science and Technology, Moscow, Russia
[2] A. A. Kharkevich Institute for Information Transmission Problems, Moscow, Russia

## ABSTRACT

**Background:** Protein phosphorylation is the best studied post-translational modification strongly influencing protein function. Phosphorylated amino acids not only differ in physico-chemical properties from non-phosphorylated counterparts, but also exhibit different evolutionary patterns, tending to mutate to and originate from negatively charged amino acids (NCAs). The distribution of phosphosites along protein sequences is non-uniform, as phosphosites tend to cluster, forming so-called phospho-islands.

**Methods:** Here, we have developed a hidden Markov model-based procedure for the identification of phospho-islands and studied the properties of the obtained phosphorylation clusters. To check robustness of evolutionary analysis, we consider different models for the reconstructions of ancestral phosphorylation states.

**Results:** Clustered phosphosites differ from individual phosphosites in several functional and evolutionary aspects including underrepresentation of phosphotyrosines, higher conservation, more frequent mutations to NCAs. The spectrum of tissues, frequencies of specific phosphorylation contexts, and mutational patterns observed near clustered sites also are different.

## INTRODUCTION

Protein post-translational modifications (PTMs) are important for a living cell (*Schweiger & Linial, 2010*; *Kurmangaliyev, Goland & Gelfand, 2011*; *Studer et al., 2016*; *Huang et al., 2017*). By changing physico-chemical properties of proteins, PTMs affect their function, often introducing novel biological features (*Pearlman, Serber & Ferrell, 2011*). To date, hundreds of thousands of PTMs in various organisms have been identified and various databases containing information about PTMs have been compiled (*Ptacek & Snyder, 2006*; *Huang et al., 2017*).

Protein phosphorylation is likely both the most common and the best studied PTM (*Ptacek & Snyder, 2006*; *Schweiger & Linial, 2010*; *Huang et al., 2017*). Phosphorylation introduces a negative charge and a large chemical group to the local protein structure, hence strongly affecting the protein conformation (*Pearlman, Serber & Ferrell, 2011*; *Nishi, Shaytan & Panchenko, 2014*). As a result, diverse cellular signaling pathways are based on

Corresponding author
Mikhail Moldovan,
mika.moldovan@gmail.com

sequential phosphorylation events (*Moses & Landry, 2010*; *Pearlman, Serber & Ferrell, 2011*; *Ardito et al., 2017*). In eukaryotes, phosphorylation sites (phosphosites) are mainly represented by serines, threonines, and tyrosines (which we here refer to as STY amonoacids), with only a minor fraction involving other amino acids, such as histidine (*Fuhs & Hunter, 2017*; *Huang et al., 2017*).

Phosphosites are overrepresented in intrinsically disordered regions (IDRs) of proteins, that is, in regions devoid of tertiary structure, usually located on the surface of a protein globule (*Iakoucheva, 2004*). Hence, studies of the evolution of phosphosites have mainly concentrated on sites located in IDRs (*Kurmangaliyev, Goland & Gelfand, 2011*; *Miao et al., 2018*). In particular, it has been shown, that phosphosites tend to arise from negatively charged amino acids (NCAs) more frequently than their non-phosphorylated counterparts, and, in a number of cases, retain structural features initially maintained by NCAs (*Kurmangaliyev, Goland & Gelfand, 2011*; *Miao et al., 2018*). As phosphorylation is often highly conserved (*Macek et al., 2007*), experimental limitations on the number of model species with established phosphosites may be overcome in evolutionary studies by formally assigning phosphorylation labels to homologous sites (*Kurmangaliyev, Goland & Gelfand, 2011*; *Huang et al., 2017*). However, this approach requires a degree of caution when dealing with evolutionary trees of substantial depths, for example, only a small fraction of yeast phosphosites are conserved between species separated by ~1,400 My (million years), while about a half of phosphosites are conserved at a shorter time (~360 My) (*Studer et al., 2016*). At smaller distances, this method may be applied to infer some evolutionary properties of phosphosites, for example, in *Drosophila* or in vertebrate species, phosphosites tend to mutate to NCA (*Kurmangaliyev, Goland & Gelfand, 2011*; *Miao et al., 2018*).

Phosphorylation can be both a constitutive modification and a way to transiently modify the protein function (*Landry et al., 2014*). In the former case, the change of a phosphosite to NCA should not cause a significant fitness reduction, as physico-chemical properties are not strongly affected, whereas in the latter case a mutation would have dire consequences (*Moses & Landry, 2010*; *Landry et al., 2014*).

In proteins, phosphosites often form co-localized groups called phosphorylation islands or phosphorylation clusters, and about a half of phosphorylated serines and threonines are located in such clusters (*Schweiger & Linial, 2010*). While individual phosphosites function as simple switches, phospho-islands are phosphorylated in a cooperative manner, so that the probability of a phosphorylation event at a focal site strongly depends on the phosphorylation of adjacent sites, and when the number of phosphorylated amino acids exceeds a threshold, the cumulative negative charge of the phosphate groups introduces functionally significant changes to the protein structure (*Landry et al., 2014*).

Accurate procedures for the identification of phosphosites and next-generation sequencing technologies yielded large numbers of well-annotated phosphosites (*Altenhoff et al., 2017*; *Huang et al., 2017*; *The UniProt Consortium, 2018*) enabling us to develop an accurate automatic procedure for the identification of phosphosite clusters we call phospho-islands. We show that clustered phosphosites exhibit evolutionary properties distinct from those of individual phosphosites, in particular, an enhanced mutation rate to

NCA and altered mutational patterns of amino acids in the phosphosite vicinity. Our study complements earlier observations on the general evolutionary patterns in phosphosites with the analysis of mutations in non-serine phosphosites and the demonstration of differences in the evolution of clustered and individual phosphorylated residues.

## MATERIALS AND METHODS

### Data

The phosphosite data for human, mouse and rat proteomes were downloaded from the iPTMnet database (*Huang et al., 2017*). The phosphorylation breadth values for the mouse dataset were obtained from *Huttlin et al. (2010)*. Human, mouse and rat proteomes were obtained from the UniProt database (*The UniProt Consortium, 2018*). Vertebrate orthologous gene groups (OGGs) for human and mouse proteomes were downloaded from the OMA database (*Altenhoff et al., 2017*). Then, all paralogous sequences and all non-mammalian sequences were excluded from the obtained OGGs.

### Alignments and trees

We searched for homologous proteins in three proteomes with pairwise BLASTp alignments (*Altschul et al., 1990*). Pairs of proteins with highest scores were considered closest homologs. The information about closest homologs was subsequently used to predict phosphosites conserved between human and rat or human and mouse which we hereinafter refer to as HMR phosphosites. OGGs were aligned by the ClustalO multiple protein alignment (*Sievers et al., 2014*) while the HMR phosphosites were identified based on Muscle pairwise protein alignments (*Edgar, 2004*). The mammalian phylogenetic tree was obtained from Timetree (*Kumar et al., 2017*).

### Phosphorylation retention upon mutations

After the identification of homologous protein pairs in human/mouse and mouse/rat proteomes and the proteome alignment construction, we identified homologous phosphosites as homologous STY residues which were shown to be phosphorylated in both species. We have shown that phosphorylation is retained on S-T and T-S mutation by comparing two pairs of retention probabilities (Fig. 1C): $p(pS–pS)$ with $p(pS–pT \mid S)$ and $p(pT–pT)$ with $p(pS–pT \mid T)$ (analogously for the phosphorylation of tyrosines), $p(pX–pX)$ being defined as the fraction of X amino acids phosphorylated in both considered species:

$$p(pX-pX) = \frac{\#(pX - pX)}{\#(pX - pX) + \#(pX - X)}$$

and $p(pX_1–pX_2)$, as the fraction of phosphorylated $X_1$ residues in one species given that in another species another amino acid residue ($X_2$) is also phosphorylated:

$$p(pX_1 - pX_2 \mid X_1) = \frac{\#(pX_1 - pX_2)}{\#(pX_1 - pX_2) + \#(pX_1 - X_2)}$$

$$p(pX_1 - pX_2|X_2) = \frac{\#(pX_1 - pX_2)}{\#(pX_1 - pX_2) + \#(X_1 - pX_2)}$$

Homologous phosphosite lists from the human/mouse and human/rat pairs were merged to produce HMR phosphosite list of human phosphosites.

## False-positive rates of phosphorylation identification by homologous propagation

We assessed the quality of the phosphorylation prediction via homologous propagation approaches by counting false-positive rates of phosphosite predictions in species with large phosphosite lists. As the numbers of predicted phosphosites drastically differed between species (*Huang et al., 2017*), we considered multiway predictions in each case as characteristics of the procedure performance. Hence, considering mouse phosphosites predicted by homology with known human phosphosites, we also considered human phosphosites predicted based on known mouse phosphosites. The false-positive rate was assessed as the proportion of incorrectly predicted phosphosites among the STY amino acids in one species homologous to phosphosite positions in other considered species.

When assessing the quality of phosphosite predictions based on phosphosites experimentally identified in at least two species, we considered human, mouse and rat and the lists of phosphosites homologous between human and mouse and between human and rat. In these cases, predictions were made for rat and mouse, respectively with the false-positive rate assessed by the same approach as in the previous case.

## Mutation matrices

To obtain single amino acid mutation matrices, we first reconstructed ancestral states with the PAML software (*Yang, 2007*). For the reconstruction, we used OGG alignments which did not contain paralogs and pruned mammalian trees retaining only organisms contributing to corresponding OGG alignments. The alignment of both extant and reconstructed ancestral sequences and the corresponding trees were then used to construct mutation matrices, where we distinguished the phosphorylated and non-phosphorylated states of STY amino acids. Here, the phosphorylation state was assigned to STY amino acids using the phosphorylation propagation approach described above. When calculating the mutation matrix, we did not count mutations predicted to happen on branches leading from the root to first-order nodes, as PAML did not reconstruct them well without an outgroup (*Koshi & Goldstein, 1996*; *Yang, 2007*). Tree pruning and calculating the mutation matrix count were implemented in ad hoc python scripts using functions from the ete3 python module (http://etetoolkit.org/).

## Disordered regions and identification of phospho-islands

Intrinsically disordered protein regions are defined here, following (*Xue et al., 2010*), as regions of proteins lacking stable and well-defined three-dimensional structure. IDRs were predicted with the PONDR VSL2 software with default parameters (*Xue et al., 2010*). This algorithm was selected, firstly, as one of the best IDR predictors yielding results highly consistent with other top-IDR predictors (*Peng & Kurgan, 2012*; *Zhou et al., 2020*), and,

secondly, as the one efficiently predicting long IDRs (*Peng & Kurgan, 2012*), which is essential for the present study.

Phosphorylated amino acids were divided into those located in predicted IDRs and those located in ordered regions (ORs). On the HMR set construction, phosphosites with conflicting IDR/OR labels were excluded from the analysis. In the analyses of separate IDR/OR mutations, we considered IDR and OR labels of amino acids to be conserved along the mammalian tree and hence inferred the remaining extant and ancestral IDR/OR states from homology with both mouse and human ORs and IDRs. We consider the premise of conserved mammalian IDRs justified here, as it is known that protein tertiary structure elements, including IDRs, are evolving slowly (*Chen et al., 2006*; *Toth-Petroczy et al., 2008*).

Phospho-islands were identified by a hidden Markov model (HMM) built upon the distributions of distances between clustered and individual phosphosites. For that, the most likely clustered/individual phosphosite assignments were obtained with the Viterbi algorithm that is guaranteed to maximize the posterior probability (*Viterbi, 1967*). The emission probabilities for the HMM were obtained as the ratio of density values in the decomposition of the distribution of amino acid distances between adjacent phosphosites in IDRs, $S$ (the likelihood ratio normalized to 1) (Fig. 2A). To select the optimal transitional probability values, we performed a stability check by analyzing the dependance of the fraction of clustered phosphosites on the transitional probability values (Fig. S1). The percentages of clustered phosphosites turned out to be extremely stable with respect to transitional probability values if the latter were smaller than 0.3. Hence, the transitional probabilities were set to 0.2 (Fig. 2B).

## Phosphosite contexts

We employed the list of phosphosite contexts as well as the binary decision-tree procedure to define the context of a given phosphosite from *Villen et al. (2007)*. The procedure is as follows. (i) Proline context is assigned if there is a proline at position +1 relative to the phosphosite. (ii) Acidic context is assigned if there are five or six E/D amino acids at positions +1 to +6 relative to the phosphosite. (iii) Basic context is assigned if there is a R/K amino acid at position −3. (iv) Acidic context is assigned if there are D/E amino acids at any of positions +1, +2 or +3. (v) Basic context is assigned if there are at least two R/K amino acids at positions −6 to −1. Otherwise, no context is assigned and we denote this as the "O" (other) context. We consider tyrosine phosphosites separately and formally assign the with the "Y" (tyrosine) context.

## Local mutation matrices

We computed local substitution matrices (LSMs) as the substitution matrices for amino acids located within a frame with the radius $k$ centered at a phosphorylated serine or threonine. When computing LSMs, we did not count mutations of or resulting in STY amino acids to exclude the effects introduced by the presence and abundance of phospho-islands. We have set $k$ to 1, 3, 5, and 7 and selected 3 as for this value we observed

the strongest effect, that is, obtained the largest number of mutations with frequencies statistically different from those for non-phosphorylated serines and threonines.

## Statistics

When comparing frequencies, we used the $\chi^2$ test if all values in the contingency matrix exceeded 20 and Fisher's exact test otherwise. To correct for multiple testing, we used the Bonferroni correction with the scaling factor set to 17 for the substitution vector comparison and to $17 \times 17$ for the comparison of substitution matrices with excluded STY amino acids. A total of 95% two-tailed confidence intervals shown in figures were computed by the $\chi^2$ or Fisher's exact test. The significance of obtained Pearson's correlation coefficients was assessed with the F-statistic.

## Code availability

Ad hoc scripts were written in Python. Graphs were built using R. All scripts and data analysis protocols are available online at https://github.com/mikemoldovan/phosphosites.

# RESULTS

## Conserved phosphosites

As protein phosphorylation in a vast majority of organisms has not been studied or has been studied rather poorly (*Huang et al., 2017*), the evolutionary analyses of phosphosites typically rely on the assumption of absolute conservation of the phosphorylation label assigned to STY amino acids on a considered tree (*Kurmangaliyev, Goland & Gelfand, 2011*; *Miao et al., 2018*). Thus, if, for instance, a serine is phosphorylated in human, we, following this approach, would consider any mutation in the homologous position of the type S-to-X to be a mutation of a phosphorylated serine to amino acid X (Fig. 1B). However, the comprehensive analysis of yeast phosphosites has shown low conservation of the phosphorylation label at the timescales of the order 100 My and more (*Studer et al., 2016*). Thus, we have considered only orthologous groups of mammalian proteins, present in the OMA database (*Altenhoff et al., 2017*). The mammalian phylogenetic tree is about 177 My deep (*Kumar et al., 2017*), which corresponds to about 50% of the phosphorylation loss in the 182 My-deep yeast *Saccharomyces-Lachancea* evolutionary path (*Studer et al., 2016*). The tree contains three organisms with well-studied phosphoproteomes: human (227,834 sites), mouse (92,943 sites), and rat (24,466 sites) (*Huang et al., 2017*) (Fig. 1A).

Still, the expected 50% of mispredicted phosphosites could render an accurate evolutionary analysis impossible. This could be partially offset by considering phosphosites conserved in well-studied lineages. Thus, we compiled a set of human phosphosites homologous to residues phosphorylated also in mouse and/or rat, which we will further refer to as human-mouse/rat (HMR) phosphosites. The HMR set consists of 53,437 sites covering 54.6% and 61.2% of known mouse and rat phosphosites, respectively, which is consistent with the above-mentioned observation about 50% phosphorylation loss in yeast on evolutionary distances similar to the ones between the human and rodent lineages (Figs. 1A and 1B).

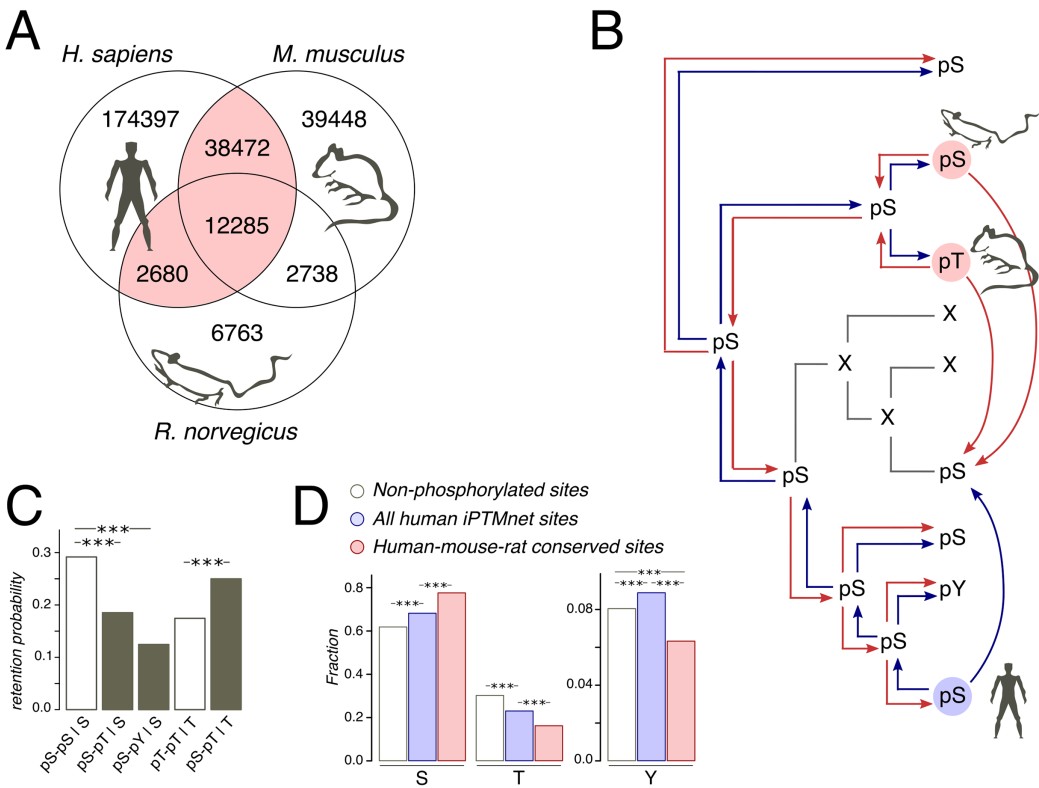

**Figure 1 Phosphosites considered in the study.** (A) Venn diagram of iPTMnet human, mouse and rat phosphosites. Intersections correspond to conserved phosphosites. The HMR phosphosite dataset is shown in pink. (B) Phosphosite assignment procedures. Given a tree of a mammalian orthologous gene group and a column in the respective alignment, we assign phosphorylation labels to ancestral and extant amino acids, firstly, by propagating labels from one species to all other species in the tree (shown as separate red and blue arrows) and, secondly, by propagating labels predicted both in the selected species (e.g., human, as shown) and in one of the remaining species (mouse and rat); this corresponds to blue and red arrows entering a given node in the tree. Phosphosites obtained by the latter procedure are referred to as the HMR phosphosite dataset. In both procedures, phosphorylation is considered to be retained both for direct and indirect STY-to-STY mutations. (C) Retention of phosphorylation upon mutation. Bars represent the probability of a conserved modification for the human dataset in the case of mutation and if mutation has not occurred. The letter after the vertical bar is an amino acid over which the probability was normalized. Three asterisks represent $p < 0.001$ ($\chi^2$ test). (D) STY amino acid content of three groups of phosphosite datasets.                                 

We consider the sites predicted by homology with the HMR set to be enriched in accurately identified phosphosites, as retaining only conserved phosphosites we substantially reduce the number of mispredictions. If we simply propagated human phosphorylation labels to mouse and *vice versa* we would get about 77.6% and 42.3% of false positive labels, respectively. However, sites conserved between human and rat or sites conserved between rat and mouse would yield about twofold lesser percentages of 41.9% and 19.9% of false positives in mouse and human, respectively. The obtained percentages can be considered as upper estimates of false positive rates, as current experimental phosphosite coverage in mammals cannot guarantee the identification of all conserved phosphosites (*Huang et al., 2017*). Thus, the HMR dataset is sufficiently robust for the prediction of phosphorylation labels in less-studied mammalian lineages.

Treatment of STY amino acids homologous to phosphorylated ones as phosphorylated yields another possible caveat, stemming from the possible loss of phosphorylation upon STY-to-STY mutations. To assess this effect, we compared the probabilities of phosphosite retention upon pSTY-to-STY mutation, pSTY indicating the phosphorylated state, and the respective probabilities in the situation when a mutation has not occurred for a pair of species with well-established phosphosite lists, that is, human and mouse (Fig. 1C). We have observed only a minor, insignificant decrease of the probabilities of the phosphorylation retention in the cases of pS-pT and pS-pY mismatches relative to the pT-pT states in mouse and human, indicating the general conservation of the phosphorylation label upon amino acid substitution. An interesting observation here is that the pS-pS states appear to be the most conserved ones (Fig. 1C). Taken together, these results indicate the evolutionary stability of phosphorylation states upon mutation.

The increased evolutionary robustness of the pS state relative to the pT and pY states should manifest as overrepresentation of phosphoserines among phosphosites with respect to non-phosphorylated amino acid positions. Thus, we assessed the relative abundancies of pSTY amino acids in the HMR dataset relative to the established human phosphosite set and to the set of non-phosphorylated STY amino acids. Serines and threonines, comprising the vast majority of the pSTY amino acids, are, respectively, over- and underrepresented in the phosphosite sets (Figs. 1C and 1D). This effect is significantly more pronounced in the HMR dataset relative to the total human phosphosite dataset, further supporting the observation about lower conservation of pT relative to pS, as the HMR dataset is enriched in conserved phosphosites by design.

## Phosphorylation islands

The distribution of distances between phosphosites is different from that of randomly chosen serines and threonines even accounting for the tendency of phosphosites to occur in intrinsiacally disordered regions (IDRs) (*Schweiger & Linial, 2010*) (Fig. 2A). However, this observation depends on an arbitrary definition of phosphorylation islands as groups of phosphosites separated by at most four amino acids (*Schweiger & Linial, 2010*). We have developed an approach that reduces the degree of arbitrariness in the definition of phospho-islands which is based on a statistical model of the distances between phosphorylated residues in phospho-islands and for individual phosphosites.

We consider only phospho-islands located in IDRs, due to two reasons. First, IDR phosphosites, being more abundant, yield reliable statistics. Second, ordered regions are largely non-uniform in terms of local structural features, for example, being localized in the protein hydrophobic core or at the surface (*Van der Lee et al., 2014*). This would render construction of the null model of between-phosphosite distances impossible without considering all protein structures of the mammalian proteome, which is currently not feasible. Hence, we will hereinafter refer to phospho-islands located in IDRs simply as phospho-islands and to non-clustered phosphosites located in IDRs as individual phosphosites.

Let $S$ be the distribution of amino acid distances between adjacent phosphosites in IDRs. The logarithm of $S$ is not unimodal (Fig. 2A), and we suggest that it is a superposition of

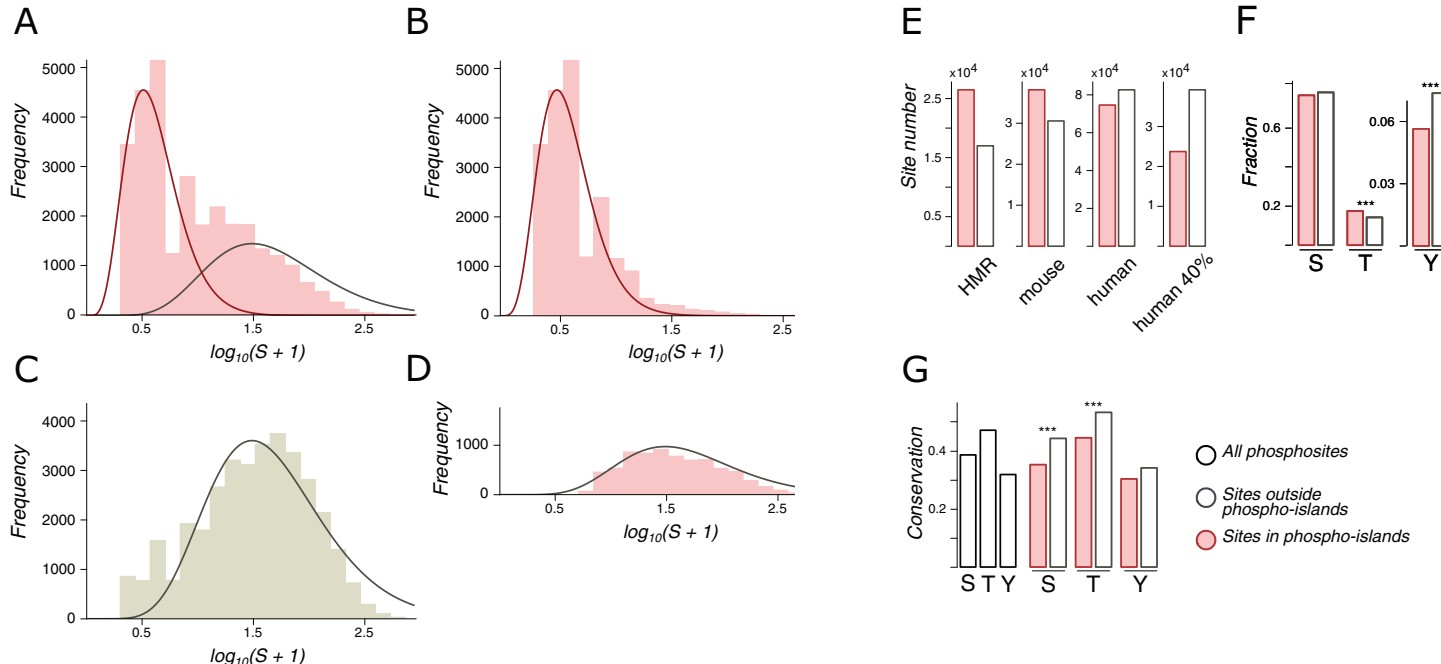

**Figure 2 Phospho-islands for the HMR phosphosite dataset.** (A) The distribution of $log_{10}(S+1)$ values (pink histogram) and its decomposition in two gamma distributions: the one for phospho-islands (red curve) and for individual phosphosites (red curve). (B) The distribution of $log_{10}(S+1)$ values for phosphosites predicted to be in phospho-islands. (C) $log_{10}(S+1)$ values for non-phosphorylated STY amino acids randomly sampled from DRs with the same sample size and amino acid content as in the HMR dataset. (D) $log_{10}(S+1)$ values for predicted individual phosphosites. (E) Numbers of individual phosphosites and sites in phospho-islands for four datasets. (F) Amino acid content of phospho-islands and individual phosphosites. (G) Frequency of mutations for phosphosites and individual amino acids. Asterisks depict significantly different values ($p < 0.001$, $\chi^2$ test).

two distributions: one generated by phosphosites in phospho-islands and the other reflecting phosphosites outside phospho-islands (left and right peaks, respectively). The latter distribution can be obtained from random sampling from IDRs of non-phosphorylated STY amino acids while preserving the amino acid composition and the sample size, as we expect individual phosphosites to emerge independently while maintaining the preference towards IDRs (Fig. 2C). Gamma distribution has a good continuous fit to log($S$+1) for randomly sampled STY amino acids located in IDRs. Given its universality and low number of parameters (*Friedman, Cai & Xie, 2006*; *Reiss, Facciotti & Baliga, 2007*; *Mendoza-Parra et al., 2013*), we have selected gamma distribution as a reasonable model for log($S$+1) (Fig. 2C). Assuming that the distribution of log($S$+1) values for phosphosites located in phospho-islands should belong to the same family and fixing the parameters of the previously obtained distribution, we decomposed the distribution of log($S$+1) values into the weighted sum of two gamma distributions, one of which corresponds to STYs located in phospho-islands and the other one, to remaining STYs in IDRs (Fig. 2A, red and gray curves, respectively). From these two gamma distributions we obtained parameters for a hidden Markov model, which, in turn, was used to map phosphorylation islands. The distributions of $S$ values for phosphosites in identified islands and the distribution for other phosphosites yielded a good match to the expected ones (Figs. 2B and 2D).

Both for the HMR and mouse datasets, more than half of phosphosites are located in phospho-islands (61% and 56%, respectively) (Fig. 2E; Figs. S2A and S2B). For human phosphosites, however, we see a larger proportion of sites (53%) located outside phospho-islands. In the latter case the distributions in the decomposition differ less, compared to the former two cases (Fig. 2A; Fig. S2). It could be caused by a larger density of phosphosites in IDRs of the human proteome, resulting from higher experimental coverage; that would lead to generally lower $S$ values, which, in turn, could cause the right peak in the $\log(S+1)$ distribution to merge with the left peak, rendering the underlying gamma-distributions less distinguishable. To validate this explanation, we randomly sampled 40% of human phosphosites, so that the sample size matched the one for mouse phosphosites; however, the results on this rarefied dataset did not change (Fig. 2E; Fig. S2C) indicating that our procedure is robust with respect to phosphosite sample sizes. Hence, phospho-islands for the human dataset are identified with a lower accuracy than those for the HMR and mouse datasets. This could be caused by different experimental technique applied to the human phosphosites, compared to the one used for mouse and rat phosphosites, and by a possibly large number of false-positive phosphosites in the former case (*Huttlin et al., 2010*; *Bekker-Jensen et al., 2017*; *Xu et al., 2017*) (see "Discussion").

In phospho-islands, the overall pSTY-amino acid composition differs from that of individual phosphosites, mainly because the fraction of threonines is significantly higher in phospho-islands at the expense of the lower fraction of tyrosines (Fig. 2F). Also, the conservation of residues in phospho-islands is larger than that of the individual sites (Fig. 2G). Overall, the general properties of clustered phosphosites seem to differ from those of individual phosphosites.

A similar attempt to decompose the $S$ distribution for phosphosites located in ordered regions yielded the distribution of $\log(S+1)$ values highly skewed to the left (small distances), even relative to the distribution of $\log(S+1)$ values in phospho-islands in IDRs (Fig. S2E). This precluded decomposition of the $S$ distribution into a weighted sum of two distributions. A more complex model possibly incorporating features of the tertiary protein structure might be required to infer and analyze phospho-islands located in ORs, which is beyond the scope of the present study.

## Mutational patterns of phosphorylated amino acids

Next, we have reconstructed the ancestral states for all mammalian orthologous protein groups not containing paralogs and calculated the proportions of mutations $P(X_1X_2)$, where $X_1$ and $X_2$ are different amino acids. We treated phosphorylated and non-phosphorylated states of STY amino acids as distinct states. We then introduced a measure of difference in mutation rates for phosphorylated STY and their non-phosphorylated counterparts. For a mutation of an STY amino acid $X$ to a non-STY amino acid $Z$ we define $R(X, Z) = P(pX \rightarrow Z)/P(X \rightarrow Z)$. If $X^*$ is another STY amino acid, $R(X, X^*) = P(pX \rightarrow pX^*)/P(X \rightarrow X^*)$. Thus, the $R$ value for a given type of mutations is the proportion of the considered mutation of a phosphorylated STY amino acid among other mutations normalized by the fraction of respective mutations of the non-phosphorylated STY counterpart. The $R$ values are thus not affected by differences in

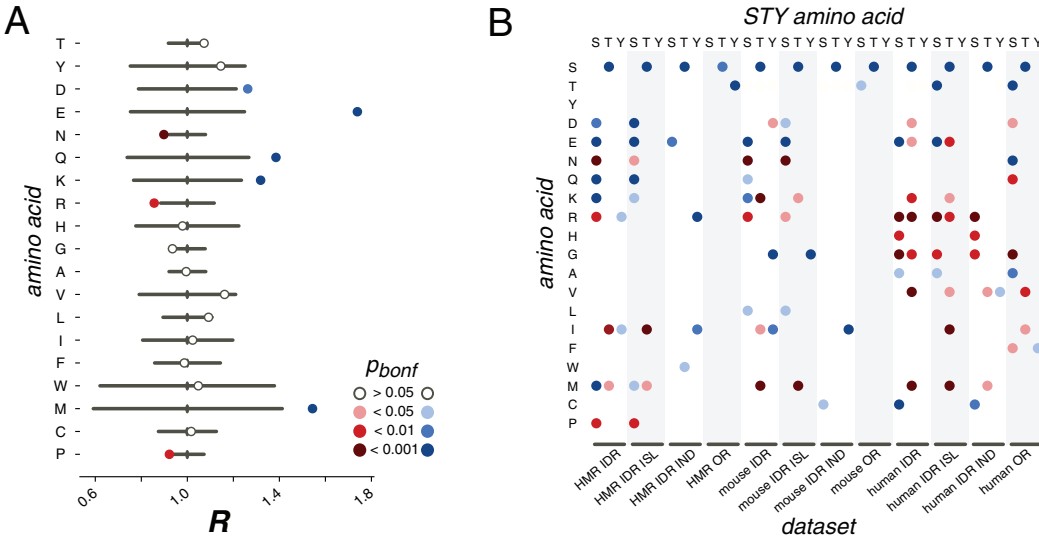

**Figure 3 $pX_0 \to X_1$ substitution vectors.** (A) $R$ values of the $pS \to X$ substitutions for serines from the HMR dataset located in DRs. (B) Substitution probabilities for phosphorylated STY amino acids significantly different from those for non-phosphorylated STY amino acids for several datasets. The significance levels are shown with the colors introduced in the panel in (A). Abbreviations on the horizontal axis: ISL, phosphosites located in phospho-islands; IND, individual phosphosites; DR, phosphosites from disordered regions; OR, phosphosites from ordered regions.

the mutation rates between phosphorylated and non-phosphorylated amino acids, as all probabilities are implicitly normalized by the mutation rates of $pX$ and $X$.

We firstly consider phosphosites located in IDRs. For phosphorserines from the HMR dataset we confirm earlier observations: phosphoserines mutate to NCA more frequently than non-phosphorylated serines (Fig. 3A). The $R$ values for serine mutation to aspartate, $R(S,D)$, and glutamate, $R(S,E)$, are both significantly larger than 1 (1.2, $p < 0.01$ and 1.7, $p < 0.001$, respectively; $\chi^2$ test) and, interestingly, they differ substantially ($p < 0.001$, multiple random Poisson sampling test). Similarly, asparagine and glutamine $R$ values differ, with $R(S,N) = 0.9$ ($p < 0.001$, $\chi^2$ test), significantly lower than 1, and $R(S,Q) = 1.4$ ($p < 0.001$, $\chi^2$ test), significantly higher than 1. The rate of mutation to lysine significantly differs for phosphorylated and non-phosphorylated serines ($p < 0.001$, $\chi^2$ test). Interestingly, the mutation rate to another positively charged amino acid, arginine, is significantly lower than expected ($p < 0.01$, $\chi^2$ test). For non-polar amino acids generally no significant differences in the $R$ values between phosphorylated and non-phosporylated serines are observed, but for methionine and proline, the calculated values are significant: $R(S,M) > 1$ ($p < 0.001$, $\chi^2$ test) and $R(S,P) < 1$ ($p < 0.01$, $\chi^2$ test).

In earlier studies, only mutations of serines or to serines had been considered, as the available data did not allow for statistically significant results for threonine and tyrosine (*Kurmangaliyev, Goland & Gelfand, 2011*; *Miao et al., 2018*). Here, we see that phosphorylated threonines from the HMR dataset tend to mutate to serines (Fig. 3B). At that, phosphorylated serines mutate to threonines more frequently than their non-phosphorylated counterparts for all considered samples, that is, for the human, mouse and HMR sets (Fig. 3B; Figs. S3–S8). Phosphorylated tyrosines tend to avoid mutations

to isoleucine ($p < 0.05$, $\chi^2$ test) and, for human samples, to arginine ($p < 0.05$, $\chi^2$ test) and glycine ($p < 0.001$, $\chi^2$ test) (Fig. 3B; Figs. S3–S8). Phospho-tyrosines in the mouse dataset show a weaker tendency for the avoidance of the mutations to aspartate than the non-phosphorylated ones ($p < 0.05$, $\chi^2$ test) while the rate of pY-to-I mutations is higher (Fig. 3B).

Separate analysis of mutations in phospho-islands and in individual phosphosites yields three observations. Firstly, alterations of mutation patterns of phosphoserines and phosphothreonines (pST) in IDRs relative to non-phosphorylated ST in IDRs are similar to the patterns observed for the clustered pST and, to a lesser extent, to those observed for individual pSTs (Fig. 3B). This is mostly due to the fact that the mutational patterns of clustered pSTs generally differ from those of their non-phosphorylated counterparts to a greater extent than the mutational patterns of individual phosphoserines do (Fig. 3B; Fig. S1). Secondly, for phosphotyrosines, alterations in their mutational patterns brought about by phosphorylation are mostly explained by individual phosphotyrosines. The mutational patterns of individual sites deviate from the ones observed for non-phosphorylated tyrosines more than those of clustered phosphotyrosines (Fig. 3B; Figs. S3–S8). Also, if we compare the $R$ values calculated for all possible mutations in clustered vs. individual phosphosites, the $R$ value corresponding to the S-to-E mutation will be significantly higher for the set of clustered phosphosites ($p = 0.009$, $\chi^2$ test, Fig. S10). Hence, we posit that the general phosphosite mutational pattern alterations can be explained mostly by mutations in clustered phosphosites for phosphoserines and phosphothreonines and by individual sites when phosphotyrosines are considered.

We also studied mutation patterns in ordered regions (ORs), and observed that phosphothreonines located in ORs demonstrate higher T-to-S mutation rates (Fig. 3B) relative to those of non-phosphorylated threonines located in ORs. Also, sites located in ORs demonstrate enhanced S-to-T and Y-to-T mutation rates relative to non-phosphorylated serines and threonines in ORs, respectively (Fig. 3B).

## Phosphosite contexts

Sequence contexts of phosphosites generally fall into three categories: acidic (A), basic (B), and proline (P) motifs, with tyrosine phosphosites comprising a special class (Y) (*Villen et al., 2007*; *Huttlin et al., 2010*). For each phosphosite from each dataset we have identified its context. As in previous studies (*Villen et al., 2007*; *Huttlin et al., 2010*), phosphosites not assigned with any of these context classes were considered as having "other" (O) motif. We studied the distribution of these motifs for all classes of phosphosites.

In IDRs, relative to ORs, we observed a higher percentage of phosphosites with assigned contexts (Fig. 4A). P-phosphosites demonstrate the highest overrepresentation in IDRs, with 25% of IDR phosphosites having the proline motif. Phospho-islands, compared to individual phosphosites, contain more phosphosites with assigned motifs relative to individual phosphosites. In IDRs, there are more B- and P-phosphosites and fewer A-phosphosites among clustered sites than among individual ones. Notably, the fraction of phospho-tyrosines is substantially higher in ordered regions. However, this effect could be

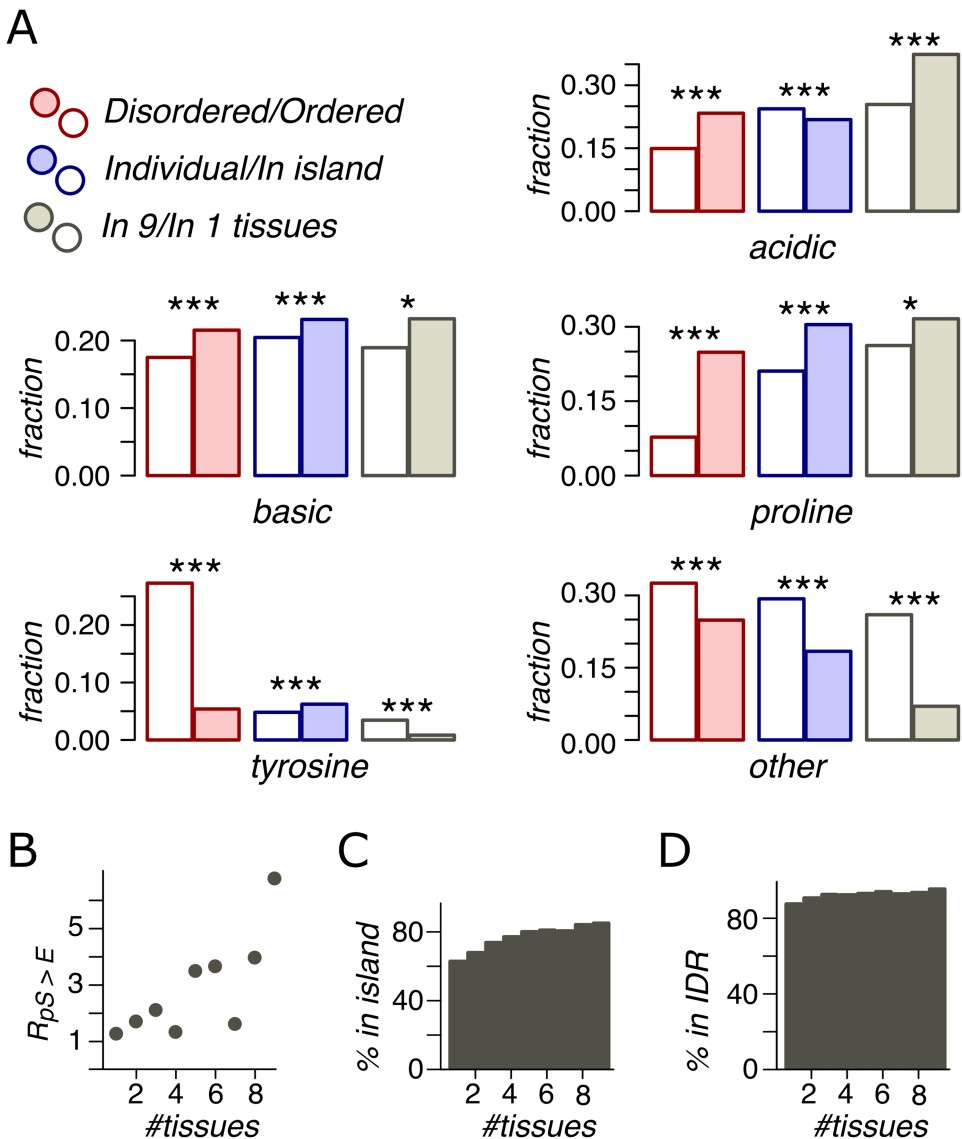

**Figure 4 Phosphosite contexts and phosphorylation breadth.** (A) Overrepresentation of phosphosite contexts in ordered vs. disoredred regions, in phospho-islands vs. individual phosphosites and for broadly vs. narrowly distributed phosphorylated amino acids. One asterisk and three asterisks indicate statistical significance at the levels of 0.05 and 0.001 respectively ($\chi^2$ test). (B) The dependance of $R$(pS,E) on the phosphosite breadth. Pearson's $r^2$ is equal to 0.53 with the $t$-test $p = 0.016$. (C) The dependance of phosphosite fraction in phospho-islands on the phosphorylation breadth ($p = 9*10^{-41}$, $\chi^2$ test). (D) Percent of phosphosites in disordered regions vs. phosphosite breadth ($p = 4.1*10^{-10}$, $\chi^2$ test).

at least partially explained by the general tendency of aromatic residues, including tyrosine, to occur in ordered protein regions (*Receveur-Bréchot et al., 2005*).

## Phosphorylation breadth

An important feature of a phosphosite is its "phosphorylation breadth", that is, the number of tissues where it is phosphorylated. In this study, the maximal phosphorylation breadth is nine, as the phosphorylation data for nine mouse tissues are available

(*Huttlin et al., 2010*). Among broadly expressed phosphosites (present in all nine tissues), compared to tissue-specific ones (present in only one tissue), very few sites have unassigned contexts (O) and almost none are tyrosine phosphosites. The fraction of acidic phosphosites (24%) is substantially lower among tissue-specific sites relative to broadly phosphorylated ones (37%) ($p < 0.001$, $\chi^2$ test) (Fig. 4A).

As mentioned above, the pS-to-E mutation yields the highest value, $R(S,E)$ (Fig. 3A) and represents the only mutation with significantly different $R$ values in phospho-islands and individual sites ($p = 0.009$, $\chi^2$ test, Fig. S1). At that, $R(S,E)$ significantly increase with increasing breadth of expression (Fig. 4B), from $R(S,E) = 1.14$ for tissue-specific phosphosites to $R(S,E) = 6.64$ for broadly expressed phosphosites ($p = 0.016$, $t$-test).

Finally, we compared percentages of phosphosites with different breadths in ORs vs. IDRs and in phospho-islands vs. individual phosphosites (Figs. 4C and 4D). As the phosphorylation breadth increases, so does the fraction of clustered phosphosites, reaching 85% for sites phosphorylated in nine tissues; the fraction of phosphosites in IDRs also increases, reaching 95.4%.

Hence, broadly expressed phosphosites have well-defined motifs, tend towards disordered regions and to phospho-islands, have mostly acidic context, and mutate to NCA more frequently than tissue-specific phosphosites.

## Mutation patterns in the proximity of phosphosites

We now show that not only phoshosites require special motifs (*Huttlin et al., 2010*), but the mutational context of clustered phosphosites differs from that of individual sites. To assess evolutionary dynamics associated with phosphosite motifs, we analyzed mutational patterns in ±3 amino acid windows of HMR ST phosphosites located in IDRs and compared them with those of non-phosphorylated ST amino acids from IDRs. The ±3 window was selected, as it yielded the strongest effect in terms of the number of mutations with rates statistically distinct from the expected ones (Figs. S11A and S11B). We did not consider phosphotyrosines, as they have not been shown to possess any discernible general motif apart from the phosphorylated tyrosine itself (*Huttlin et al., 2010*).

We introduce the measure $Q$ defined as $Q(X_1^p \rightarrow X_2^p) = P(X_1^p \rightarrow X_2^p)/P(X_1^n \rightarrow X_2^n)$, where $X_1^p$ and $X_2^p$ are amino acids near phosphorylated serines and threonines and $X_1^n$ and $X_2^n$ are amino acids near non-phosphorylated serines and threonines. $Q$ measures overrepresentation of a given mutation in the proximity of pST amino acids relative to ST amino acids. We also considered sites located in phospho-islands and individual phosphosites separately (Fig. 5; Figs. S11C and S11D).

In the whole HMR dataset, 22 types of non-phosphorylated amino acid substitutions out of the total of 289 have $Q$ values statistically different from the expected value 1 ($p < 0.05$, $\chi^2$ test with the Bonferroni correction), among them three pairs of mutually reverse mutations (Fig. 5). As expected from the conservation of the phosphosite contexts, mutations between positively charged amino acids and NCAs, potentially changing acidic to basic contexts and *vice versa*, are underrepresented, whereas E-to-D, D-to-E and K-to-R,

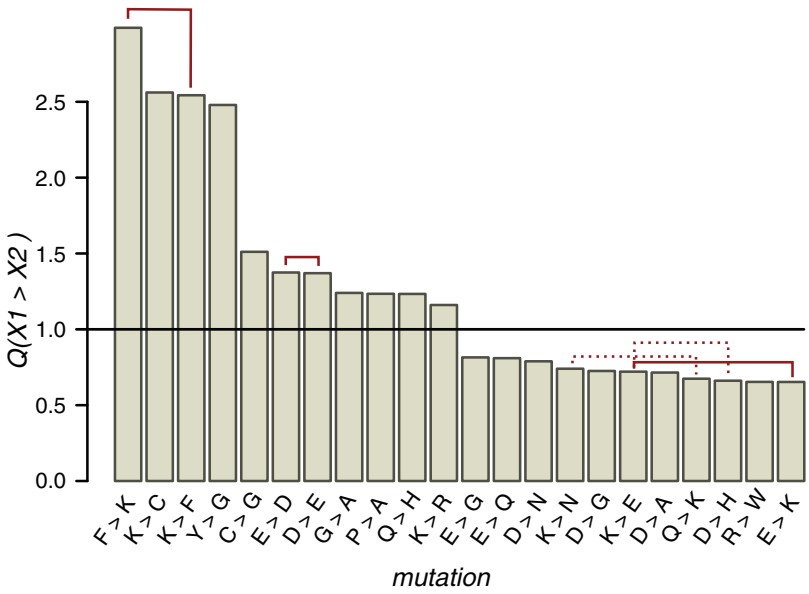

**Figure 5 Q values of mutations near ST phosphosites with probabilities significantly different from the expected ones.** Solid red lines connect mutually reverse mutations. Dashed lines indicate quazy-reverse mutations of amino acids with common chemical properties.

not changing the context type, are overrepresented. The P-to-A substitution is overrepresented, thus indicating the instability of proline contexts. Interestingly, all three mutations with $Q$ values exceeding 2.5 involve lysine, two of them being reverse mutations F-to-K and K-to-F. The fourth most overrepresented mutation, Y-to-G with $Q$(Y→G) = 2.5, could explain the lack of tyrosine phosphosites in IDRs, as a large fraction of IDR phosphosites are clustered with the distances between sites not exceeding three amino acids. Thus, a large $Q$(Y→G) value would lead to general underrepresentation of tyrosines in IDRs.

Types of mutations with significant $Q$ values generally differ near clustered and individual phosphosites (Figs. S11C and S11D). E-to-D, not changing the local acidic context (*Huttlin et al., 2010*), is overrepresented and E-to-K, disrupting the acidic context (*Huttlin et al., 2010*), is underrepresented in both cases. On the other hand, around individual phosphosites, $Q$(F→K) = 3.4 and $Q$(P→A) = 1.12, indicating an enhanced birth rate of the basic context and disruption of the proline context, respectively. The R-to-D mutation, disrupting the local basic context, also is overrepresented near individual phosphosites. In general, among seven overrepresented mutations near clustered phosphosites, only the K-to-P mutation disrupts the local basic context in favor of the proline context and among seven overrepresented mutations near individual phosphosites, three mutations (E-to-F, R-to-D, and P-to-A) could be regarded as context-disrupting. Hence, the individual phosphosite contexts are somewhat less evolutionary stable and thus the lower percentage of individual phosphosites with identifiable contexts might be due to specific local context-disrupting mutation patterns for these phosphosites.

## DISCUSSION

### Clustered vs. individual phosphosites

We have demonstrated that clustered phosphosites differ from non-clustered ones in a number of aspects: (i) overrepresentation of phosphothreonines and underrepresentation of phosphotyrosines in phospho-islands (Fig. 2F); (ii) stronger conservation of clustered phosphoserines and phosphothreonines (Fig. 2G); (iii) larger proportion of sites phosphorylated in many tissues (Fig. 4C); (iv) significantly larger probability of mutations to glutamate for clustered relative to the individual phosphoserines; (v) larger fraction of sites with specific motifs in phospho-islands (Fig. 4A); (vi) mutational patterns in the proximity of phosphosites consistent with the context-retention hypothesis (Fig. 5). What are possible explanations for the observed effects?

Underrepresentation of phosphotyrosines in phospho-islands could be explained by phosphorylation of clustered phosphosites being co-operative. As serines and threonines are more similar to each other in their tendency to being phosphorylated by similar enzymes than they are to tyrosine (*Villen et al., 2007*; *Huttlin et al., 2010*; *Landry et al., 2014*; *Studer et al., 2016*), one would expect phospho-tyrosines to disrupt co-operative phosphorylation of adjacent ST amino acids by being phosphorylated independently, thus introducing a negative charge which would affect phosphorylation probabilities of the neighboring amino acids (*Landry et al., 2014*). Hence phospho-tyrosines could have been purged by selection from pST clusters.

Secondly, phosphosites located in phospho-islands are more conserved than individual ones (Fig. 2G), as opposed to an earlier hypothesis that individual phosphosites are more conserved than their clustered counterparts (*Landry et al., 2014*). Our result seems to contradict the notion that the cellular function of phosphosites in an island depends on the number of phosphorylated residues rather than specific phosphorylated sites, whereas individual phosphosites operate as single-site switches and hence should be more conserved (*Landry et al., 2014*). However, this argument implies that phosphorylation of most individual phosphosites is important for the organism's fitness, which may be not true (*Landry et al., 2014*; *Miao et al., 2018*) and hence our results do not contradict the model of evolution of functionally important phosphosites.

Overrepresentation of phosphosites with defined motifs among the clustered ones (Fig. 4A) and reduced numbers of mutations disrupting the local contexts of the clustered sites (Figs. S10C and S10D) may indicate enhanced selective pressure on clustered phosphosites and their contexts. An indirect support of this claim comes from the overrepresentation of ubiquitously phosphorylated sites among the clustered ones (Fig. 4C). Indeed, broad phosphorylation requires a stronger local context and indicates the reduced probability of a phosphosite being detected simply due to the noise inherent to the phosphorylation machinery (*Landry et al., 2014*).

Mutations of phosphoserines located in IDRs to NCA are generally overrepresented among all mutations of the type pS-to-X relative to the corresponding mutations of non-phosphorylated serines (Fig. 3B). This effect is stronger for clustered phosphosites

and for ubiquitously phosphorylated sites. Together with the observation about clustered phosphosites being on average more broadly phosphorylated than the individual ones, this suggests that a large fraction of phosphosite clusters might be phosphorylated (nearly) constitutively, and thus changes of individual phospho-serines to NCAs could experience lesser degrees of negative selection acting upon the corresponding mutations, as these mutations introduce smaller degrees of local electric charge shifts on the protein globule than the mutations of non-phosphorylated serines to NCAs do.

## Two types of mutations

In all considered phosphosite datasets, we have observed two types of pSTY-to-X mutations overrepresented relative to STY-to-X mutations (Fig. 3B): (i) pSTY-to-pSTY, especially pT-to-pS mutation and (ii) pSTY-to-NCA, especially pS-to-E mutations. The former effect could be explained by the relaxed selection against pST-to-pST mutations due to the phosphorylation machinery often not distinguishing between serines and threonines (Huttlin et al., 2010; Miao et al., 2018). The overrepresentation of pT-to-pS mutation for all datasets, including sites located in ORs, could stem from the higher probability of phosphosite retention following a pT-to-pS mutation relative to the probability of phosphorylated threonine retention when no mutations have occurred (Fig. 1C). Thus, the observed enhanced pT-to-pS mutation rate could be due to the enhanced evolutionary stability of serine phosphorylation relative to the threonine phosphorylation.

The enhanced serine-to-NCA mutation rates could stem from the physico-chemical similarity of phosphorylated serines and NCAs: both types of residues introduce negatively charged groups of similar size to the protein globule. Thus, if phosphorylation is (almost) constitutive, that is, happens very frequently in a large number of tissues, we would expect the serine-to-NCA mutation rate to be enhanced. Indeed, ubiquitous phosphorylated serines have the pS-to-E mutation rate more than six-fold larger than the S-to-E mutation rate (Fig. 4B). However, the same pattern does not hold for phospho-threonines (Fig. 3B).

The differences in the mutation rates observed for phosphosites are stronger when clustered phosphosites are considered. Although this might be explained by individual phosphosites likely resulting from noise generated by the phosphorylation machinery (Landry et al., 2014), this could also indicate a general pattern of phosphosites constantly arising at random points of the proteome due to a constant evolutionary process. If phosphorylation at a focal site turns out to be advantageous, its individual context could be reinforced yielding broader phosphorylation pattern of this site or, alternatively, other phosphosites could emerge in the vicinity of this phosphosite, thus forming phospho-islands. As the vast majority of broadly phosphorylated sites are clustered, and clustered phosphosites demonstrate stronger phosphosite-specific features than individual phosphosites do, we suggest that formation of phosphorylation clusters around beneficial phosphosites is the prevalent process compared to context reinforcement of just one site. However, this hypothesis requires further verification.

## Human phosphosites

The results obtained for the human set of phosphosites differ somewhat from those for the mouse and HMR sets, like in cases with different STY amino acids representation among phosphorylated amino acids (Fig. 1D), proportion of phosphosites located in phospho-islands (Fig. 2E) or some mutational patterns of phosphorylated STY amino acids (Fig. 3B). This could be explained by differences in experimental procedures used to obtain phosphosite lists for human and for mouse and rat. Whereas for classic laboratory organisms, phosphosites are obtained directly from the analysis of an organism or an analysis of its live organ (Huttlin et al., 2010), for human phosphosite inference immortalized cell lines, such as HeLa, are used (Bekker-Jensen et al., 2017; Xu et al., 2017), with conditions differing from those in vivo, and hence one could expect different patterns of phosphorylation. In particular, the lower rate of mutations to NCA could be explained by overrepresentation of sites with noisy phosphorylation manifesting only in cell lines under the conditions of experiments. The mutation of such a residue to NCA would most likely result in the deleterious effect of an average non-phosphorylated serine mutation to NCA (Jin & Pawson, 2012). Thus, we propose that phosphosites conserved between human and rodent lineages, called here HMR sites, are more robust with respect to experimental techniques, and hence are better suited for phosphosite evolutionary studies.

## Evolution of non-studied phosphosite groups

Previous studies dedicated to the evolution of phosphosites have focused on phosphoserines located in IDRs. The large datasets employed in the present study enabled us to assess the patterns of phosphothreonines, phosphotyrosines and sites located in ORs. Apart from the largely enhanced pT-to-pS mutation proportions relative to T-to-S ones (Fig. 3B) no patterns with straightforward biological explanation were observed in these cases. However, an interesting observation here is the consistent, significantly enhanced rate of pY-to-I mutations relative to the Y-to-I mutations in the mouse and HMR datasets (Fig. 3B).

## Perspectives

We propose a simple yet accurate homology-based approach for the ancestral phosphosite inference yielding in our case the set of HMR phosphosites. As the predicted fractions of phosphorylation labels falsely assigned to internal tree nodes are much smaller than the ones for other phosphosite datasets, HMR set poses a valuable source of data for evolutionary studies. A practical extension of our homology-based approach could be a phosphosite prediction procedure incorporating additional pieces of information such as the tendency of phosphosites to cluster, the local phosphosite contexts, and the tree structure into the probabilistic model, which would predict phosphosites with a high degree of accuracy. On the other hand, it would be interesting to infer the interplay between phosphorylation and selection using population-genetics data.

### Funding

This study was supported by the Russian Foundation of Basic Research under grant 18-29-13011. The funders had no role in study design, data collection and analysis, decision to publish, or preparation of the manuscript.

### Grant Disclosures

The following grant information was disclosed by the authors:
Russian Foundation of Basic Research: 18-29-13011.

### Competing Interests

Mikhail S. Gelfand is an Academic Editor for PeerJ.

### Author Contributions

- Mikhail Moldovan conceived and designed the experiments, analyzed the data, prepared figures and/or tables, authored or reviewed drafts of the paper, and approved the final draft.
- Mikhail S. Gelfand conceived and designed the experiments, authored or reviewed drafts of the paper, and approved the final draft.

### Data Availability

All scripts and data analysis protocols potentially relevant for the reproduction of the present study is available at GitHub: https://github.com/mikemoldovan/phosphosites.

### Supplemental Information

Supplemental information for this article can be found online at http://dx.doi.org/10.7717/peerj.10436#supplemental-information.

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
