# Peer review of "Phospho-islands and the evolution of phosphorylated amino acids in mammals"

_PeerJ, doi:10.7717/peerj.10436_

## Round 0.1 · original submission · Major Revisions

Please address all the critiques of both reviewers and revise your manuscript accordingly.

Reviewer 1 ·

Basic reporting

Manuscript is well written and referenced. It is also logically structured and well focused.

A few small issues I noticed:

line 109 and others – HMR used several times before definition

line 120 – ratention → retention
line 147 – mouse. → mouse

line 232 - “We consider the HMR set to be enriched in accurately predicted phosphosites”, wording conflicts with my understanding; the previous paragraph describes this set as the union of intersections between human-mouse and human-rat experimental annotations. These are then used to MAKE predictions by transfer (as stated in the last sentence of the paragraph), but are not themselves predictions.

Fig 1C,D – Data exceed axis

Fig 2A,B,C,D – no axis, no ticks, some small illegible text on B and D. A bit over stylized in my opinion, conventional axes for C and D would be more clear

Paragraph beginning 291 – incorrect references to supplementary figure 8

line 319 – No supplementary figure S8E

Supplementary Figures – two figures named S9

Experimental design

Generally good; selected datasets, computational methods, and analysis are appropriate.

One issue with design: It is not clear why DR phosphosites are used for island modelling but OR phosphosites are excluded. This is not justified in the text. This division seems to complicate interpretation somewhat, since disorder and island are entangled in the model. Why not model all phosphosites, then compare with disorder in the analysis.

Also, please clarify how DRs are defined, particularly when comparing homologs (do both need to be disordered, just one?)

Finally, line 423, Are non-phosphorylated ST amino acids required to be in DRs as well? Not clear, but it seems that they should be to control for DR associated amino acid bias.

Validity of the findings

In general, conclusions are clear and supported by data.

This reviewer is a bit confused by finding related to predicted phosphosite islands throughout the manuscript. Are these only phosphosite islands predicted in DRs? Please clarify.

Related, the statement on line 315 is troubling: “We do not observe phospho-islands in ordered regions.” This seems inaccurate since the data seems to suggest the opposite, since the distance distribution matches islands in DRs. (1) This would not be an issue if all phosphosites were modeled, then compared with DRs. (2) An unexplored implication of this is that single sites are much more common in DRs than ORs, this seems unexpected and worth exploring, or at least discussing.

A bit of speculation, VSL2B is fairly sensitive to aromatics, including tyrosine, are OR islands enriched in Y relative to DR islands?

Reviewer 2 ·

Basic reporting

The manuscript is well written and thoroughly referenced throughout.

Acronyms should be defined explicitly at the first occurrence (HMR, HMM, etc)

Some spelling errors throughout (residues, amino acids etc) – please check this

In Fig 2,3,5 axes labels should be clarified. Figure 5 caption can be made more detailed.

Experimental design

The manuscript discusses the phenomenon of non-uniform distribution of phosphosites along protein sequences. The authors present an interesting evolutionary analysis to identify and characterize the properties of phosphorylation clusters (termed phospho-islands). While there is thorough comparison and discussion of the results in the context of existing literature, new results could be emphasized more clearly. In particular, can the authors extend their discussion of the implications of their findings related to the evolutionary patterns in phosphosites more generally?

Validity of the findings

The analysis is comprehensive and statistically sound. I believe two aspects of the data analysis should be addressed further. Specifically:

1) Prediction of disordered regions should be discussed in more detail. The authors should note the diversity of predictive methods for IDPs/IDRs. Would the variability in IDP definitions and prediction methods alter the results observed? How do the authors define DR and OR here? For varied definitions, would results still be statistically significant? In particular, as related to Figure 4 and associated discussion.

2) Notably, the authors should motivate why they use the Viterbi method (from 1967) for calculating phosphorylation clusters. Again, how sensitive are the results to the model?

---

## Round 0.2 · accepted · Accept

All critiques were adequately addressed and the manuscript was amended accordingly. Therefore, I am please to accept it for publication in PeerJ.